# Thrombospondin-4 controls matrix assembly during development and repair of myotendinous junctions

Arul Subramanian, Thomas F Schilling*

Department of Developmental and Cell Biology, University of California, Irvine, Irvine, United States

**Abstract** Tendons are extracellular matrix (ECM)-rich structures that mediate muscle attachments with the skeleton, but surprisingly little is known about molecular mechanisms of attachment. Individual myofibers and tenocytes in *Drosophila* interact through integrin (Itg) ligands such as Thrombospondin (Tsp), while vertebrate muscles attach to complex ECM fibrils embedded with tenocytes. We show for the first time that a vertebrate thrombospondin, Tsp4b, is essential for muscle attachment and ECM assembly at myotendinous junctions (MTJs). Tsp4b depletion in zebrafish causes muscle detachment upon contraction due to defects in laminin localization and reduced Itg signaling at MTJs. Mutation of its oligomerization domain renders Tsp4b unable to rescue these defects, demonstrating that pentamerization is required for ECM assembly. Furthermore, injected human TSP4 localizes to zebrafish MTJs and rescues muscle detachment and ECM assembly in Tsp4b-deficient embryos. Thus Tsp4 functions as an ECM scaffold at MTJs, with potential therapeutic uses in tendon strengthening and repair.

## Introduction

Cellular structure and function depend on dynamic interactions with extracellular matrix (ECM) proteins, defects in which cause many diseases such as muscular dystrophies and osteoarthritis (*Emery, 2002*; *Mayer, 2003*; *Kanagawa and Toda, 2006*; *Carmignac and Durbeej, 2012*; *Maldonado and Nam, 2013*). Tendons and ligaments are especially rich in ECM proteins, predominantly laminins (Lams) and collagens (Cols) (*Hauser et al., 1995*; *Kannus, 2000*; *Kjaer, 2004*; *Södersten et al., 2007*; *Snow and Henry, 2009*; *Schweitzer et al., 2010*; *Aparecida de Aro et al., 2012*; *Charvet et al., 2011*). These multimeric proteins assemble into extremely strong fibrillar structures capable of resisting the contractile forces of muscles and enabling movement (*Banos et al., 2008*; *Thorsteinsdóttir et al., 2011*; *Thorpe et al., 2013*). Muscles interact with these tendon ECM proteins through integrin (Itg) heterodimers as well as the dystrophin-associated glycoprotein complex to form attachments at myotendinous junctions (MTJs) (*Kannus et al., 1998*; *Kardon, 1998*; *Blake et al., 2002*; *Bassett et al., 2003*; *Henry et al., 2005*; *Carmignac and Durbeej, 2012*). While the organization of the ECM at MTJs has been described (*Kardon, 1998*; *Aparecida de Aro et al., 2012*), the developmental processes underlying its establishment and maintenance are poorly studied.

In zebrafish embryos, early MTJs form as epithelial attachments between muscle fibers and ECM at somite boundaries (*Henry et al., 2005*; *Snow and Henry, 2009*). Initially this ECM is rich in fibronectin (Fn) but accumulates other ECM proteins as it matures. Like other vertebrate tendons, these early embryonic MTJs form through transmembrane interactions between ECM proteins with Itgs and the dystroglycan-complex, thereby linking the ECM with the muscle cytoskeleton (*Henry et al., 2001*; *Crawford et al., 2003*; *Hall et al., 2007*; *Câmara-Pereira et al., 2009*; *Jacoby et al., 2009*; *Goody et al., 2010*; *Charvet et al., 2011*) Downstream components of Itg signaling such as Paxillin (Pxn),

*For correspondence: tschilli@uci.edu

**eLife digest** Tendons, the tough connective tissues that link muscles to bones, are essential for lifting, running and other movements in animals. A matrix of proteins, called the extracellular matrix, connects the cells in a tendon, giving it the strength it needs to prevent muscles from detaching from bones during strenuous activities.

To achieve this strength, extracellular matrix proteins bind to one another and to receptors on the muscle cell surface that are linked to its internal scaffolding, thereby organizing other proteins into a structure called a myotendinous junction. However, despite the essential roles of tendons, scientists do not fully understand how this organization occurs, or how it can go awry.

Subramanian and Schilling screened zebrafish for genes that are essential for proper muscle attachment, and zeroed in on a gene encoding a protein called Thrombospondin-4b (Tsp4b). A similar protein helps to connect muscle and tendon cells in fruit flies. Without Tsp4b, zebrafish are able to form connections between muscles and tendons, but the muscles detach easily during movement. This weakened connection is caused by disorganization of the proteins in the extracellular matrix, which results in reduced signaling from the muscle cell receptors.

When a human form of this protein was injected into zebrafish embryos lacking Tsp4b, it settled into the junctions between muscle and tendon cells. The human protein repaired the detached muscles and restored the proper organization of the matrix. This improved the strength of the muscle-tendon attachment in the treated fish embryos, suggesting that similar injections could also help to strengthen and repair muscles and tendons in people.

Talin (Tln) and Integrin Linked Kinase (ILK) are also required for establishment and maintenance of functional MTJs in both zebrafish and mice (*Gheyara et al., 2007*; *Conti et al., 2008*; *Postel et al., 2008*; *Câmara-Pereira et al., 2009*).

The arrangement of ECM proteins at MTJs also changes in architecture and composition to withstand the forces exerted by muscle contraction (*Kjaer, 2004*; *Câmara-Pereira et al., 2009*; *Snow and Henry, 2009*; *Charvet et al., 2013*; *Bricard et al., 2014*). Lams, for example, become incorporated into collagen fibrils at MTJs, in contrast to the Lam meshwork that forms around Schwann cells and or Lam fibril networks in endothelia or lung alveolar cells (*Hamill et al., 2009*). At somite boundaries in mice, Lam–Itg interactions are essential for elongation and differentiation of muscle progenitors (*Bajanca et al., 2006*). Lam deposition (as well as localization of Itgs, FAK, Paxillin (Pxn), and Fn to MTJs) also depends on Itg signaling and Rho GTPases that regulate actin cytoskeletal dynamics (*Hamill et al., 2009*). Thus, bidirectional Itg signaling is required for MTJ maturation (*Parsons et al., 2002*; *Crawford et al., 2003*; *Snow et al., 2008*; *Snow and Henry, 2009*). In zebrafish embryos, this includes a gradual assembly of collagen fibrils between 1–6 days post fertilization into an orthogonal arrangement at myosepta (*Bader et al., 2009*; *Charvet et al., 2011*). Mutations in human LAMA2 cause a congenital form of muscular dystrophy called merosin-deficient muscular dystrophy (*Tome et al., 1994*; *Kanagawa and Toda, 2006*; *Carmignac and Durbeej, 2012*) and COL6 mutations cause Ullrich congenital muscular dystrophy (*Bertini et al., 2011*; *Bönnemann, 2011*; *Grässel and Bauer, 2013*; *Pan et al., 2013*), highlighting the importance of an appropriately organized muscle ECM. However, mechanisms that control this assembly of the MTJ matrix are largely unknown.

In *Drosophila*, the long splice form of the Itg ligand thrombospondin (Tsp)-TspA (*Chanana et al., 2007*; *Subramanian et al., 2007*), is a critical calcium-binding ECM protein secreted by tenocytes (tendon cells) that binds Itgs (Position Specific (PS)-beta and PS-alpha2 subunits) on myoblasts. Of the five vertebrate Tsp genes, subclass B including Tsp4 and Tsp5 have been observed in connective tissues associated with the musculoskeletal system (*Hauser et al., 1995*; *Tucker et al., 1995*; *Hecht et al., 1998*; *Jelinsky et al., 2010*; *Frolova et al., 2014*) and have been shown to function as homo- and hetero-pentamers through a conserved coiled-coil region (*Hauser et al., 1995*; *Narouz-Ott et al., 2000*; *Södersten et al., 2006*). Human TSP4 levels are highly elevated in patients with Duchenne muscular dystrophy (DMD), α-sarcoglycan deficiency, as well as in cardiac ECM in response to stress, and TSP5 levels increase during joint injury, osteoarthritis and cartilage degradation (*Chen et al., 2000*; *Hecht et al., 1998*; *Timmons et al., 2005*). Purified Tsp4 and Tsp5 also interact with a multitude of ECM proteins including Lam, Col and Fn in vitro (*Narouz-Ott et al., 2000*; *Chen et al., 2007*). Tsp4 facilitates collagen fibril packing in mouse tendons and

ECM of the heart (*Frolova et al., 2014*). However, there are no known functions for Tsps during formation of muscle attachments and MTJ development in vertebrates.

We have identified a novel zebrafish Tsp, Tsp4b, expressed and localized at all muscle attachment sites in the developing embryo and in adults. We show that zebrafish Tsp4b, in its pentameric form alone, interacts with ECM proteins such as Lam, activates Itg signaling within muscles, and is required for establishment and maintenance of muscle attachments. Furthermore, recombinant human TSP4 is functionally interchangeable with Tsp4b and capable of repairing and strengthening tendons when provided exogenously. Our results reveal a novel mechanism for pentameric Tsp4 in ECM protein assembly and maintenance at MTJs and a potential therapeutic approach for improving tendon strength and repair.

## Results

### Requirements for Tsp4b in muscle attachments

Through an in situ expression screen for markers of muscle attachments, we identified a zebrafish *tsp4*, *tsp4b* (69% similar to human TSP4; *Figure 1—figure supplement 1A–C*), which is expressed at all muscle attachment sites (*Figure 1—figure supplement 1E,F*). In zebrafish, two genes share sequence similarity with other vertebrate *Tsp4* genes—*Tsp4a* and *Tsp4b* (previously designated as zgc: 111910, *tsp-4a* and *thbs4* on chromosome 21). Similar to other subclass B Tsps, Tsp4b is predicted to be secreted as a pentamer as it contains a conserved $CX_2C$ motif (CQAC—Cys-Gln-Ala-Cys) identical to human TSP4 in its hydrophobic coiled-coil oligomerization domain (CCD) (25/36 residues are identical with human and mouse Tsp4), which is required for inter-subunit disulfide linkage (*Efimov et al., 1994*) (*Figure 1—figure supplement 1C*). In situ analyses revealed *tsp4b* mRNA throughout the differentiating myotomes of embryonic somites beginning at 16 hr post fertilization (hpf, *Figure 1—figure supplement 1D*). Expression disappears in myoblasts as they differentiate and by 60 hpf becomes restricted to putative tendon cells near somite boundaries and along the horizontal myoseptum (arrowheads in *Figure 1—figure supplement 1E*) as well as at all muscle attachment sites of the head by 72 hpf (arrowheads in *Figure 1—figure supplement 1F*). The expression of *tsp4b* resembles that of *tenomodulin* (*tnmd*), a known tenocyte-specific marker (*Figure 1—figure supplement 2*), and their relatives are co-expressed in tendons and ligaments in humans (*Docheva et al., 2005*; *Jelinsky et al., 2010*). Combined fluorescent localization of *tsp4b* mRNA and myosin heavy chain (MHC) protein revealed that down regulation of *tsp4b* in myotome occurs abruptly as the wave of muscle differentiation passes medio-laterally through each somite (*Figure 1—figure supplement 3A–H*; *Devoto et al., 1996*; *Henry et al., 2005*). By 60 hpf, *tsp4b* expression was only detected in putative tenocytes along the somite boundaries, where muscle attachments have been established (*Figure 1—figure supplement 3I–P*).

A polyclonal antibody raised against the unique N-terminus of zebrafish Tsp4b revealed extracellular protein localization around the notochord and medial somite boundaries at 20 hpf (*Figure 1A–C*) where myofibers of the axial musculature first elongate and attach, and this localization progressed laterally as more lateral fibers formed functional attachments at the somite boundary. By 72 hpf, Tsp4b protein was detected at the ends of all larval axial, appendicular, pharyngeal and extraocular muscles (*Figure 1D–L*). In the cranial region, these include muscle-cartilage attachments, inter-muscular attachments (between segments of the sternohyoideus [SH] muscle) and muscle-soft tissue attachments. These results suggest that during initial stages of muscle development in the trunk, myoblasts secrete their own Tsp4b to initiate attachment and MTJ assembly, and at later stages of somite muscle maturation, Tsp4b levels are maintained by secretion from tenocytes in mature MTJs.

Zebrafish embryos injected with antisense morpholino oligonucleotides (MOs) (0.32 ng/embryo) targeting the translation start site of *tsp4b* completely lacked Tsp4b protein by 72 hpf, as determined by whole-mount immunostaining with anti-Tsp4b. *tsp4b*-MO injected embryos (hereafter referred to as Tsp4b-deficient) were slightly curved downward, but showed no defects in muscle morphology or swimming ability (*Figure 2A–C*). However, muscle contractions induced with mild electrical stimulation caused dramatic muscle detachment in *tsp4b*-deficient embryos (*Figure 2D–F,J*). While stimulation with repeated 8 millisecond pulses at 30 volts caused the occasional isolated myofiber to detach in 23% (N = 16/68) of wild-type embryos, similar stimulation led to large portions of somites with detached fibers in 76% (N = 60/79) of *tsp4b*-deficient animals (*Figure 2J*). Furthermore, this phenotype was dependent both on the strength of stimulation as well as the dose of *tsp4b*-MO (*Figure 2—figure supplement 1A,B*). This weakening of muscle attachments was specific to reduction of Tsp4b since it

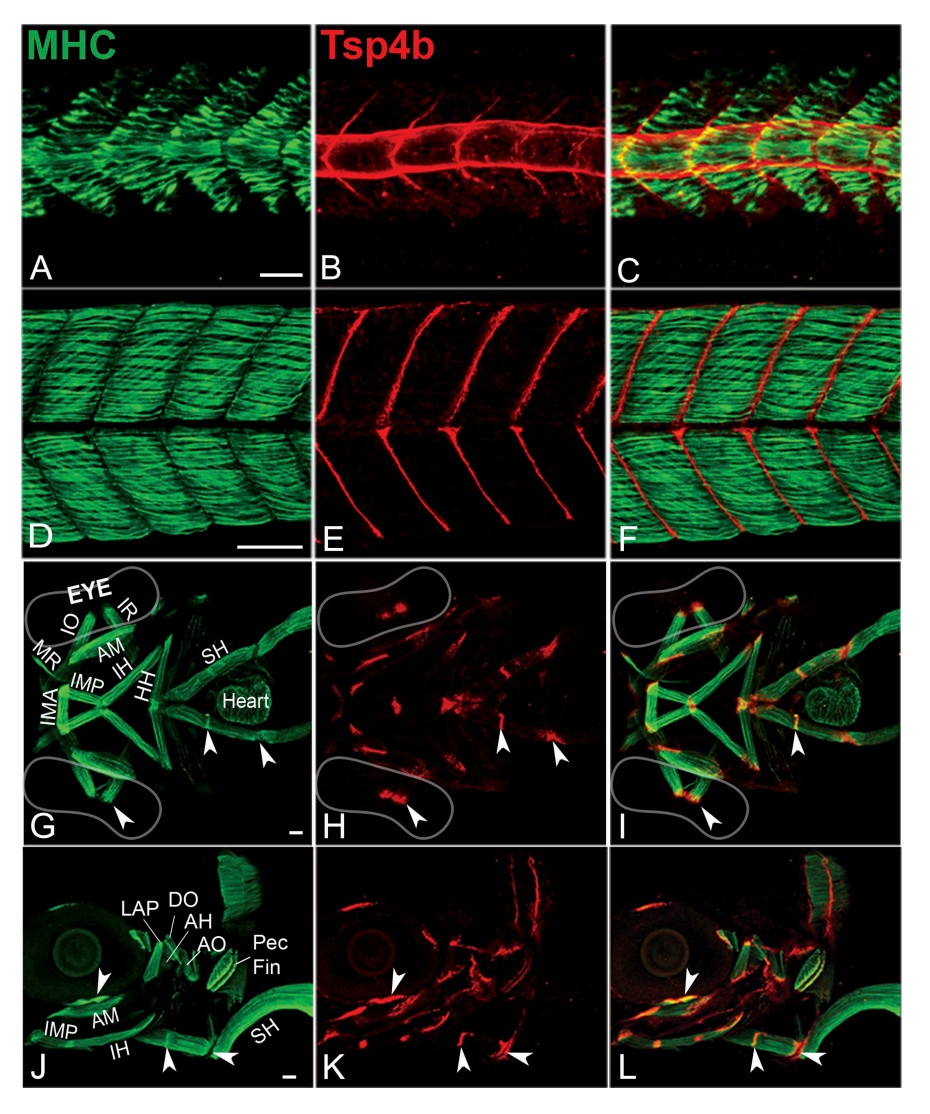

**Figure 1**. Zebrafish Tsp4b localizes to all muscle attachments. (**A–L**) Whole mount immunostaining of wild type embryos using anti-MHC (**A, D, G, J**; green) and anti-Tsp4b (**B, E, H, K**; red) and merged (**C, F, I, L**). (**A–C**) 20-22 hpf and (**D–F**) 72 hpf (lateral view) trunk showing early Tsp4b localization around notochord and medial somite boundaries (**B** and **C**) and later at somite boundaries (**E** and **F**). (**G–L**) Ventral (**G–I**) and lateral (**J–L**) views of 72 hpf showing Tsp4b at cranial muscle attachments. Abbreviations: AM-Adductor Mandibularis, AH-Adductor Hyoideus, AO-Adductor Operculae, DO-Dilator Operculae, HH-HyoHyal, IH-InterHyal, IMA-InterMandibularis Anterior, IMP-InterMandibularis Posterior, IO-Inferior Oblique, IR-Inferior Rectus, LAP-LevatorArcus Palatini, MR-Medial Rectus, SH-SternoHyoideus. Scale bar = 30 microns.

The following figure supplements are available for figure 1:

**Figure supplement 1**. Tsp4b is expressed at muscle attachments.

**Figure supplement 2**. *tsp4b* and *tnmd* are expressed in tenocytes at MTJs.

**Figure supplement 3**. Tsp4b expression is downregulated in myoblasts as they differentiate.

was partially rescued by injection of full-length *tsp4b* mRNA (*Figure 2K*). Furthermore, a mosaic distribution of exogenous mRNA restored Tsp4b protein specifically at the attachment sites of rescued myofibers in a dose-dependent manner (*Figure 2G–I*, *Figure 2—figure supplement 2*).

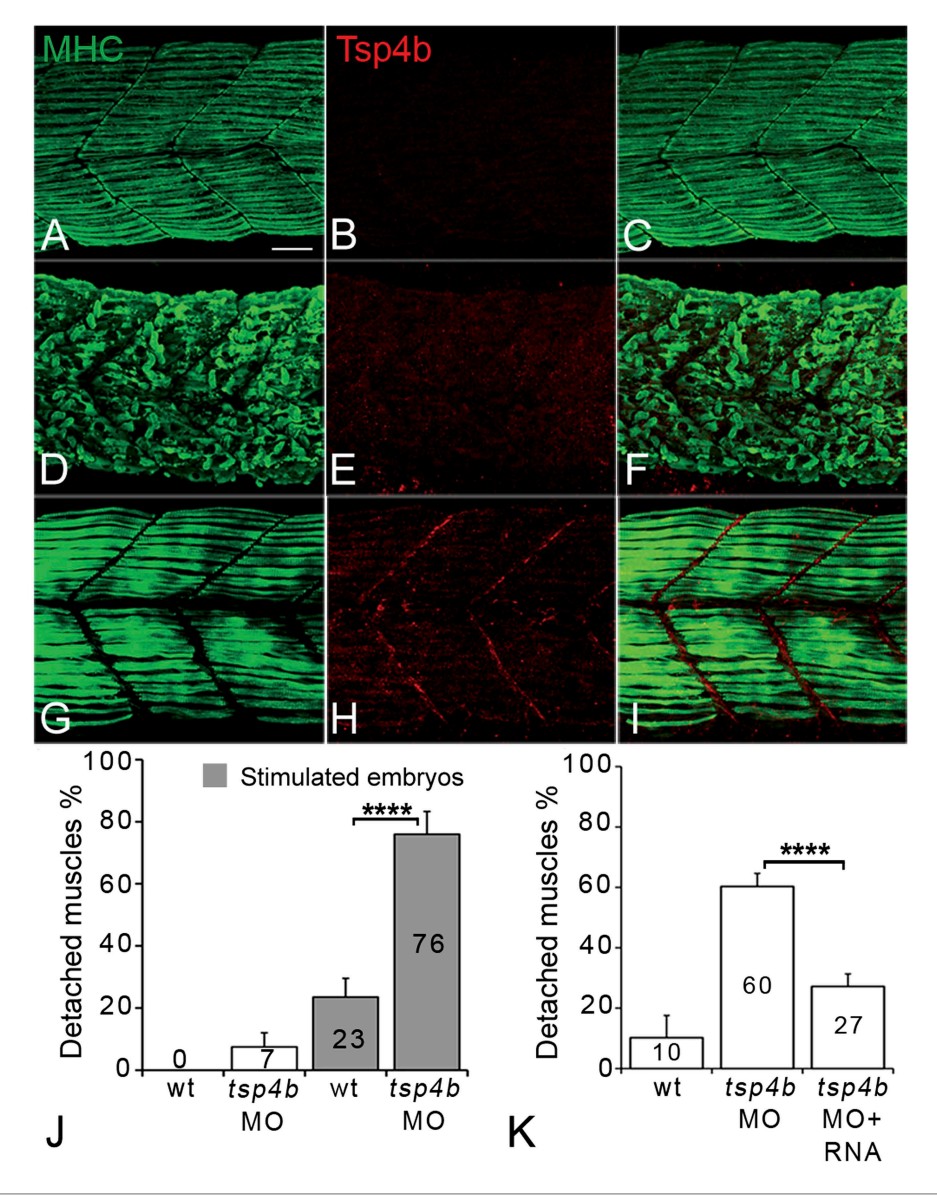

**Figure 2**. Tsp4b is required for muscle attachment. Whole mount immunostaining of 36 hpf Tsp4b-deficient embryos using anti-MHC (**A**, **D**, **G**; green) and anti-Tsp4b (**B**, **E**, **H**; red) and merged (**C**, **F**, **I**). (**A–C**) Injection of 0.32 ng *tsp4b*-MO eliminates Tsp4b protein at 72 hpf but myofibers attach. (**D–F**) Electrical stimulation (30 V) of these larvae causes muscle detachment. (**G–I**) Co-injection of *tsp4b* RNA (80 pg/embryo) rescues muscle attachment and Tsp4b localization. (**J**) Histogram showing muscle detachment in 76% (N = 79) of stimulated Tsp4b-deficient embryos (Chi squared test p-value<0.001). (**K**) Co-injection of *tsp4b* mRNA rescues muscle attachment in 67% (N = 92) of stimulated Tsp4b-deficient embryos (Chi squared test p-value<0.001). (p-value representation legend: significant *<0.05, highly significant **<0.01, extremely significant ***<0.001, extremely significant ****<0.0001). Scale bar = 30 microns.

The following figure supplements are available for figure 2:

**Figure supplement 1**. Tsp4b-deficient muscles show dose-dependent detachment upon stimulation.

**Figure supplement 2**. Exogenous *tsp4b* mRNA rescues Tsp4b localization in a dose-dependent manner.

**Figure supplement 3**. Ultrastruture of MTJs in Tsp4b deficient embryos.

In order to understand the effects of reducing Tsp4b levels on ECM organization at the MTJ, we examined MTJ ultrastructure with transmission electron microscopy (TEM). While wild-type embryos at 72 hpf showed basement membranes (BM) flanking a tightly packed electron dense MTJ (~278 nm, *Figure 2—figure supplement 3A,E*), both stimulated wild-type embryos and Tsp4b-deficient embryos prior to stimulation showed perturbation of the ECM at MTJs with intermittent separations of BM (~423 nm and ~527 nm, respectively; *Figure 2—figure supplement 3B,C,E*). In contrast, electrically stimulated Tsp4b-deficient embryos showed catastrophic defects in both ECM and BM integrity as well as a dramatic increase in MTJ separation (~2668 nm, *Figure 2—figure supplement 3D,E*), while muscle fibers remained intact. These results demonstrate a requirement for Tsp4b in maintenance and strengthening of MTJs.

## Tsp4b functions in both muscle and tendon to maintain MTJ integrity

During development of the axial musculature in zebrafish, early myofibers attach at somite boundaries by 20–24 hpf (*Devoto et al., 1996*), at least a day before tendon progenitors are detectable at somite boundaries by *tsp4b* mRNA expression (*Figure 1—figure supplement 3*). To address cell autonomous roles for Tsp4b in muscle versus tendon, we used cell transplantation to create mosaic embryos in which wild type mesoderm capable of forming either muscles or tenocytes was transplanted into Tsp4b-deficient hosts. Consistent with a requirement for Tsp4b in muscle attachment, both isolated donor-derived, wild-type muscle cells (*Figure 3A–D*), as well as putative tenocytes along somite boundaries (*Figure 3E–H*), restored Tsp4b protein and attachment of both donor and adjacent host myofibers (70%, N = 73), and these remained attached even when stimulated (*Figure 3I–L*; 40%, N = 51). This demonstrates a non cell-autonomous role of Tsp4b in maintenance of muscle attachments.

## Tsp4b is required for muscle-specific integrin signaling

Tsps function as Itg ligands. For example, interactions between vertebrate subclass A Tsps and Itgs are essential for wound healing (*Kyriakides and Bornstein, 2003*; *Bornstein et al., 2004*). The RGD (Arg-Gly-Asp) motif in *Drosophila* Tsp is required for its interactions with muscle-specific Itg to form stable attachments (*Chanana et al., 2007*; *Subramanian et al., 2007*). Zebrafish Tsp4b lacks an RGD but contains a non-canonical Itg-binding (KGD—Lys-Gly-Asp) motif in its C-terminus (*Figure 1—figure supplement 1C*; *Ruoslahti, 1996*; *Adams, 2004*). We hypothesized that secreted Tsp4b in the ECM interacts with Itgs on muscle cell surfaces to promote attachment. To address this we examined effects of depleting Tsp4b on Itg activation and functional interactions with downstream Itg signaling

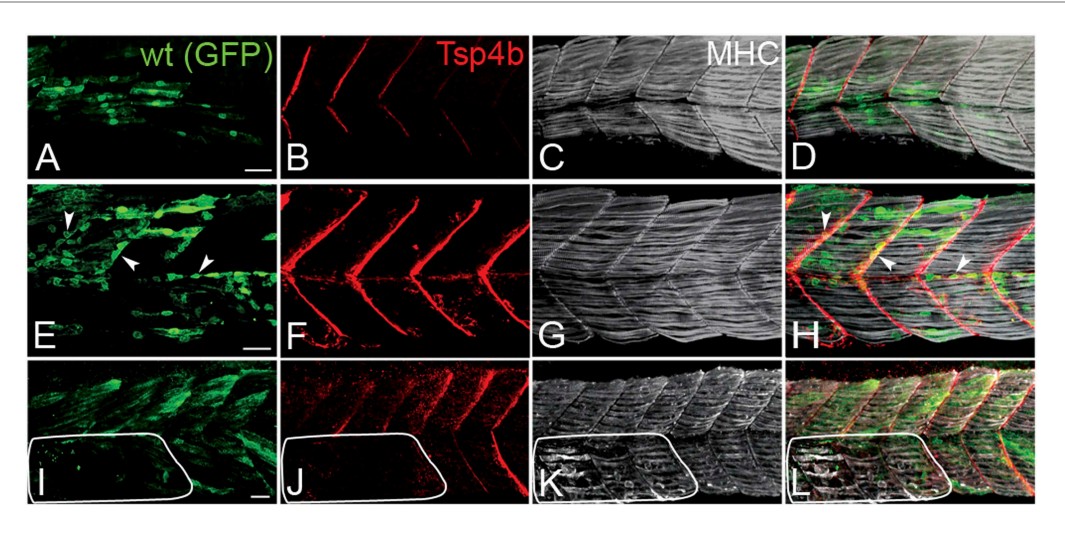

**Figure 3**. Tsp4b has a non cell-autonomous function in muscle attachments. (**A–L**) 48 hpf (lateral views) of genetic mosaics generated by cell transplantation. GFP-labeled wild type muscle cells (green) (**A** and **I**) or putative tenocytes (arrow heads) (green) (**E**) were grafted into Tsp4b-deficient host embryos and stained with anti-Tsp4b (**B**, **F**, **J**; red), and anti-MHC (**C**, **G**, **K**; gray/white). (**I–L**) Transplants locally rescued muscle detachment after stimulation in regions where Tsp4b was restored (white line denotes region lacking Tsp4b). Scale bars = 30 microns.

components. Tsp4b-deficient animals showed severe reductions in key indicators of Itg signaling within myofibers. First, in 20–22 hpf embryos injected with *tsp4b*-MO, levels of activated focal adhesion kinase (FAK, detected with an antibody that recognizes phosphorylation at Tyrosine 861—pFAK$^{y861}$) were dramatically reduced (*Figure 4A–D,G*), which was rescued by injection of *tsp4b* mRNA (*Figure 4E–G*; *Henry et al., 2001*), and at later stages (36 hpf) were mislocalized or lost in local regions along somite boundaries (*Figure 4H–K*), which was also rescued by co-injection of *tsp4b* mRNA (*Figure 4L,M*). In regions where pFAK was mislocalized along the somite boundaries, muscle attachments appeared disorganized. Second, simultaneous partial reduction of Tsp4b (sub-threshold doses of *tsp4b*-MO) and Itg signaling in heterozygous *integrin-linked kinase* (*ilk$^{+/-}$*) mutants, neither of which causes any detachment on its own, led to dramatic and widespread muscle detachment upon stimulation (*Postel et al., 2008*; *Figure 4N*). Third, in Tsp4b-deficient embryos, expression and localization of a GFP-tagged Paxillin (Pxn)—*Tg[Pxn-GFP]* - was significantly reduced, indicating a severe reduction in Itg signaling within the myofibers themselves (*Figure 4—figure supplement 1A–C*). In addition, a *Tg[Itga5-RFP]* transgenic line showed specific localization of RFP at MTJs, while Itga6-GFP localized more broadly along the entire length of each muscle fiber, which is reduced in Tsp4b-deficient embryos (*Figure 4—figure supplement 2A–D*). These results are consistent with a functional role for Tsp4b as a ligand for muscle-specific Itgs at MTJs required to stabilize muscle attachments and point to Itga5 as one likely receptor.

## Tsp4b is required for laminin localization at MTJs

In addition to Tsp4, the myotendinous ECM (both early zebrafish myosepta and MTJs) contains Lams, Dystrophin (Dys), Tenascin, Fn, Cols and other Itg ligands. Lam and Dys deficiencies cause muscular dystrophies in humans and zebrafish. *lam* mutants show defects in myoseptum formation (*lama2*) and muscle patterning (*lamb1* and *lamc1* [*Hall et al., 2007*; *Peterson and Henry, 2010*]). Many of these ECM components interact with Itgs, and the defects in Itg activation that we have observed suggest that Tsp4b may regulate assembly of some of these other ECM components (*Narouz-Ott et al., 2000*; *Frolova et al., 2012*). To address this hypothesis we examined Lam localization (with a pan-Lam antibody) in Tsp4b-deficient embryos. We found that injection of *tsp4b*-MO caused a reduction in localization of Lam at somite boundaries at 24 hpf (*Figure 5A–D,G,N*), which was rescued by co-injection of *tsp4b mRNA* (*Figure 5E–G*). This correlated with disorganized muscle attachments. At later stages (36 hpf), Tsp4b-deficient embryos showed discontinuities in Lam distribution at somite boundaries (*Figure 5H–K,N*), with many fibers forming aberrant attachments at sites of localized Lam, which could be rescued by co-injection of *tsp4b* mRNA (*Figure 5L,M*). *lam* mRNA (*lama2, lamb1, lamc1, lamc2*) levels were not significantly affected by *tsp4b* depletion suggesting that these effects are post-transcriptional (*Figure 5—figure supplement 1*) and most likely due to Tsp4b-Lam interactions. In vitro binding assays using purified rat TSP4 have shown interactions with Lam, Col, and Fn among other ECM components (*Narouz-Ott et al., 2000*). Thus, Tsp4b could interact with Lam in a similar heteromeric complex, and initiate an Itg signaling cascade necessary for maintenance of MTJs.

## Pentameric Tsp4b is required for ECM organization and Itg signaling at MTJs

The reduction of Itg signaling and Lam localization in Tsp4b-deficient embryos suggests that Tsp4b has both a functional role in muscle adhesion and attachment as well as a structural role in ECM assembly at the MTJ. Itg ligands commonly bind Itgs through a canonical RGD motif, which fails to bind when Aspartate (D) is replaced with Glutamate (E) (*Erb et al., 2001*; *Balasubramanian and Kuppuswamy, 2003*; *Takahashi et al., 2007*). Tsp4 contains a non-canonical KGD binding motif. To test requirements for this motif in promoting Itg activation in muscles, we designed a KGE mutant *tsp4b* mRNA (*Figure 6A*). Injecting the mutant form (KGD>KGE) of *tsp4b* mRNA failed to rescue the muscle detachment phenotype in Tsp4b-deficient embryos (*Figure 6B*), confirming a requirement for Itg binding in the function of Tsp4b.

Subclass B Tsps function as pentamers through the highly conserved 36 residue coiled-coil motif (*Figure 6A*). To test requirements for oligomerization of Tsp4b in its localization we perturbed Tsp4b function by preventing it from forming the functional homo-pentameric form. To do this, we mutated the conserved CCD of Tsp4b, CQAC (*Efimov et al., 1994*; *McKenzie et al., 2006*) to SQAS (Ser-Gln-Ala-Ser) to prevent formation of inter-subunit disulfide bridges that are required for pentamerization. To verify *SQAS-gfp* mRNA expression, we injected wild type *tsp4b-gfp* and *SQAS-gfp* into one-cell stage embryos and analyzed them for expression at gastrula stages. Both *SQAS-gfp* and *tsp4b-gfp* were expressed similarly and strongly fluorescent at 7–8 hpf, suggesting that the mutated SQAS protein is

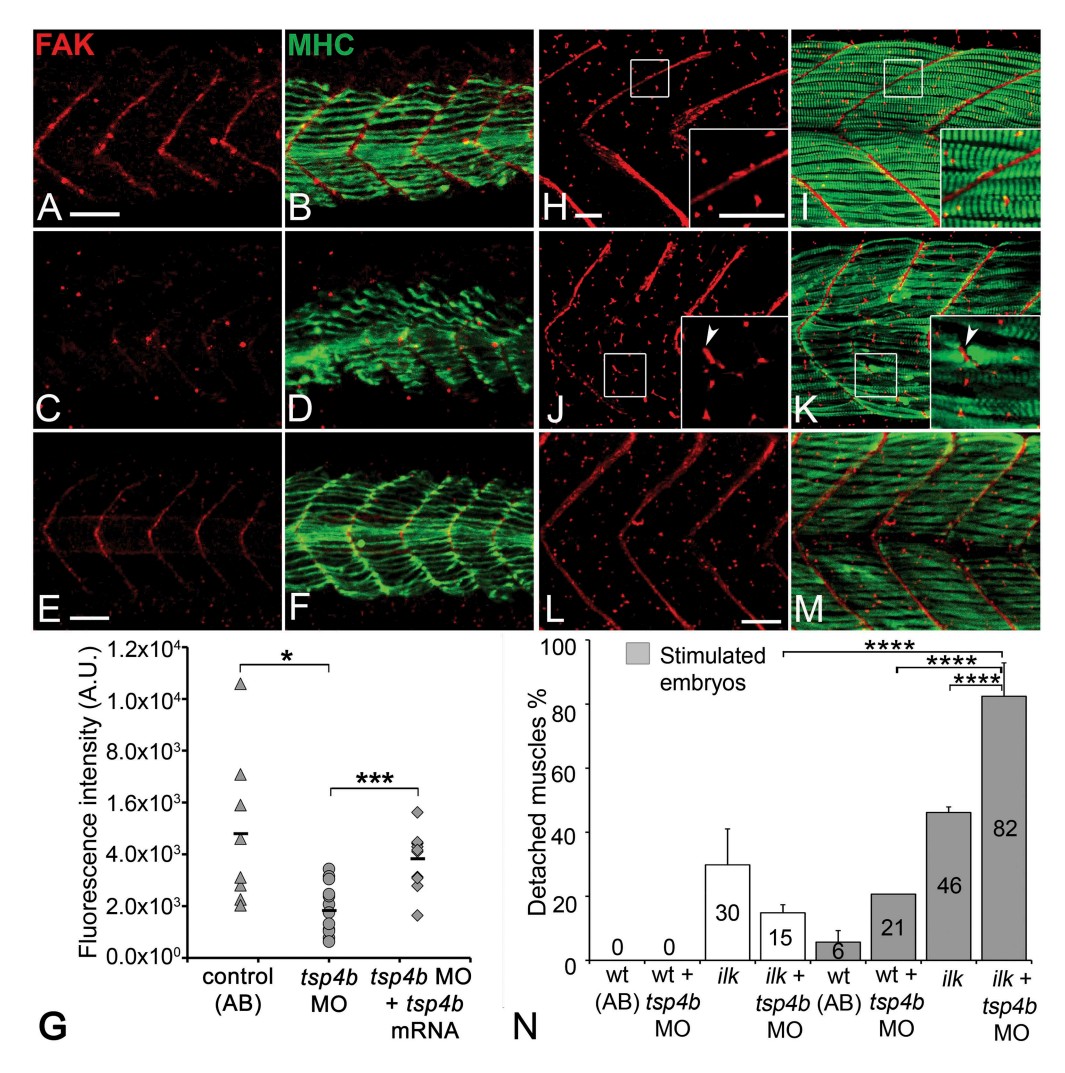

**Figure 4**. Tsp4b is required for muscle-specific integrin signaling at MTJ. (**A–F**) Lateral views of 20–24 hpf embryos stained with anti-phosphorylated (Tyrosine 861) FAK (pFAK; red) and anti-MHC (green). (**A** and **B**) pFAK localizes to the ends of early myofibers at wild-type somite boundaries. (**C** and **D**) Reduced pFAK levels in Tsp4b-deficient embryos. (**E** and **F**) pFAK levels are restored in Tsp4b-deficient embryos injected with full length *tsp4b* mRNA. (**G**) Fluorescence intensity measurements (arbitrary units [A.U.]) for pFAK staining along somite boundaries confirm significant reductions in Tsp4b-deficient embryos, and partial rescue by co-injection of full length *tsp4b* mRNA (*t* test: one tailed, unequal variance; p-value: wt and tsp4b-deficient <0.05; Tsp4b-deficient and Tsp4b-deficient + *tsp4b* RNA p<0.001). (**H–M**) 36 hpf (lateral views) stained with anti-pFAK (red), and anti-MHC (green). Insets show higher magnification images of white boxed areas. (**H** and **I**) pFAK localizes to muscle sarcolemma. (**J** and **K**) In Tsp4b-deficient embryos, pFAK is reduced/discontinuous at somite boundaries. pFAK associates with ectopic muscle attachments (arrowheads). (**L** and **M**) pFAK localization is restored in Tsp4b-deficient embryos injected with full length *tsp4b* mRNA. (**N**) Embryo percentages (N = 70 embryos) with detached muscles from an intercross between two *ilk*[+/−] heterozygotes, injected with sub-threshold amounts (0.16 ng) of *tsp4b*-MO and stimulated (30 V) (Chi squared test; p-value: wt+ *tsp4b*-MO (stimulated) and *ilk*+*tsp4b*-MO (stimulated) p<0.0001, *ilk* (stimulated) and *ilk*+*tsp4b*-MO (stimulated) p<0.0001, *ilk*+ *tsp4b*-MO and *ilk*+*tsp4b*-MO (stimulated) p<0.0001). Scale bar = 30 microns.

The following figure supplements are available for figure 4:

**Figure supplement 1**. Tsp4b-deficient muscles show reduced localization of Paxillin.

**Figure supplement 2**. Itga5-*RFP* localizes to MTJs and Itga6-GFP localizes to muscle, and both require Tsp4b.

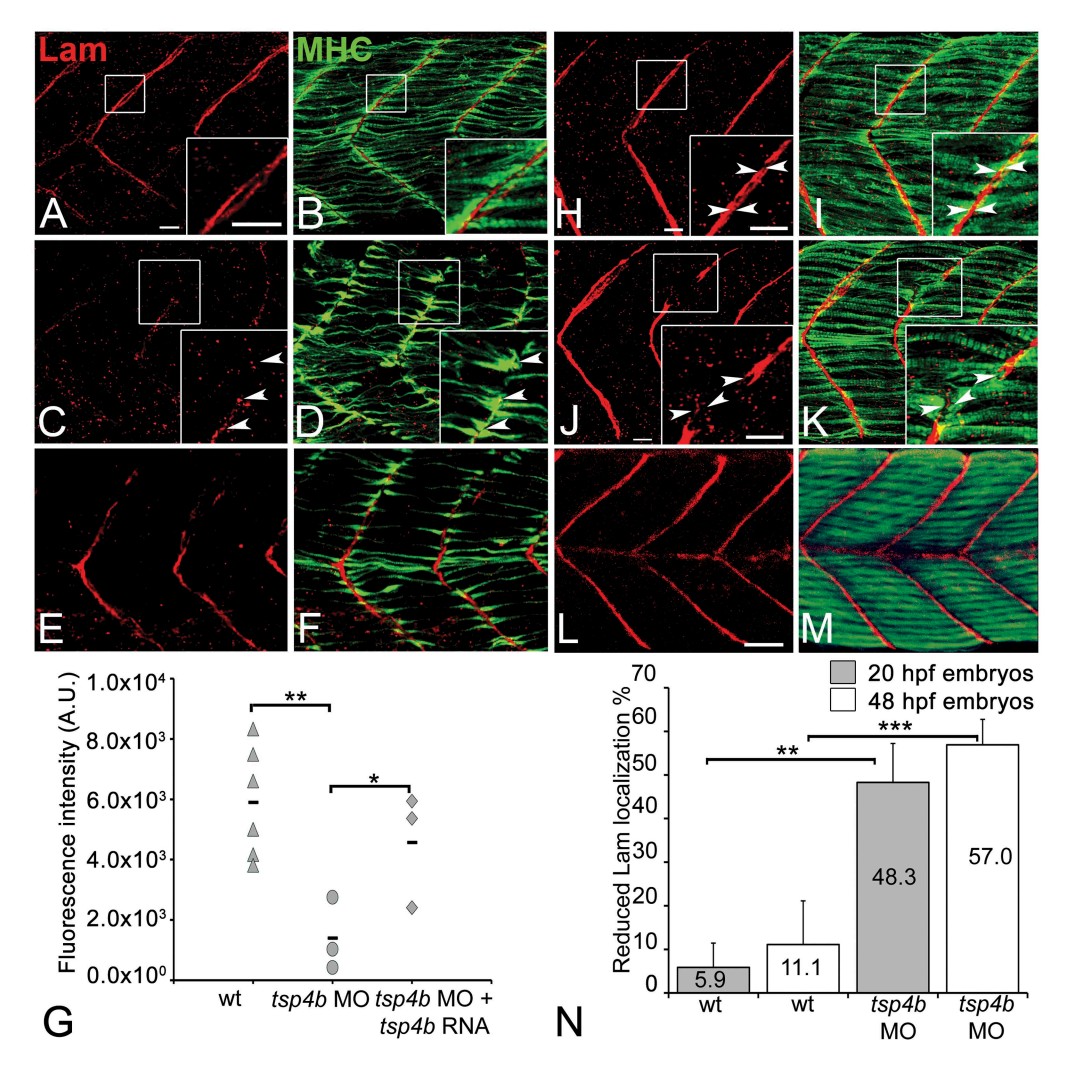

**Figure 5**. Tsp4b is required for Laminin assembly at MTJs. (**A–F**) Lateral views of somites in 20 hpf embryos stained with anti-MHC (green) and anti-pan-Laminin (Lam, red) antibodies. Insets show higher magnification images of the areas marked by white boxes, where Lam localizes to developing myotendinous ECM (arrowheads). (**A** and **B**) Wild-type siblings. (**C** and **D**) *tsp4b*-MO injected embryos showing local loss of Lam at 20 hpf, and ectopic muscle attachments (arrowheads) at remaining Lam foci. (**E** and **F**) Lam localization was restored in Tsp4b-deficient embryos injected with full length *tsp4b* mRNA. (**G**) Fluorescence intensity measurements (arbitrary units [A.U.]) at somite boundaries of anti-Lam in 20–24 hpf wild type controls versus embryos injected with tsp4b-MO or co-injected with tsp4b-MO and tsp4b RNA. (*t* test: one tailed, unequal variance; p-value: wt and Tsp4b-deficient <0.01, Tsp4b-deficient and Tsp4b-deficient and *tsp4b* RNA <0.05) Scale bar = 30 microns. (**H** and **I**) Lateral views of somites in wild type embryos at 36 hpf stained with anti-pan-Lam (red), and anti-MHC (green). Insets show higher magnification images of white boxed areas. Lam localizes to myotendinous ECM (**I**, arrowheads). (**J** and **K**) In Tsp4b-deficient embryos, Lam is reduced/discontinuous at somite boundaries (**K**, arrowheads). (**L** and **M**) Lam localization is restored in Tsp4b-deficient embryos injected with full length *tsp4b* mRNA. (**N**) Embryo percentages (20 hpf embryos N = 30, 72 hpf embryos N = 50) with reduced/mislocalized Lam at 20 and 48 hpf in wild type and Tsp4b-deficient embryos. (Chi squared test; p-values: 20 hpf **<0.01, 48 hpf ***<0.001) Scale bar = 30 microns.

The following figure supplements are available for figure 5:

**Figure supplement 1**. Tsp4b depletion does not alter *lam* transcription.

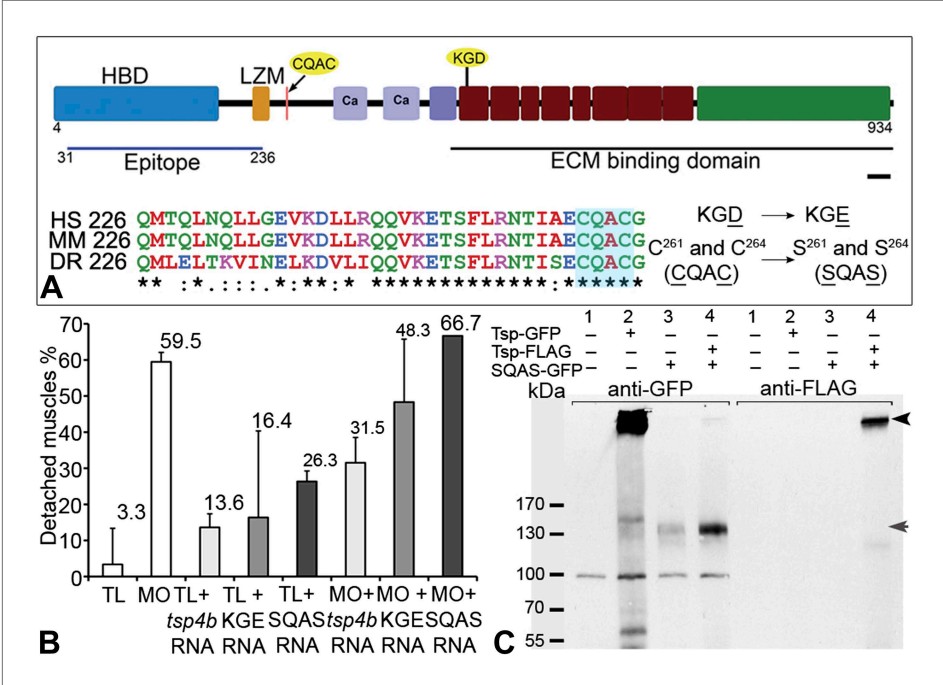

**Figure 6**. Integrin binding and pentamerization of Tsp4b are essential for its function. (**A**) Schematic representation of Tsp4b domains showing the location of the conserved coiled-coil region with its CQAC motif (70% identical across species). Amino acid substitutions used to study the functions of KGD and CQAC motifs are underlined. (**B**) Muscle detachment frequencies in uninjected wild-type controls, wildtypes injected with *tsp4b* morpholino (MO), *tsp4b* RNA, KGE *tsp4b* mutant RNA, SQAS *tsp4b* mutant RNA, or co-injected with *tsp4b* MO and *tsp4b* RNA, KGE *tsp4b* mutant RNA, or SQAS *tsp4b* mutant RNA, after stimulation (N = 60 embryos each). (**C**) Western blot performed on whole embryo protein extract using anti-GFP and anti-FLAG antibodies under non-reducing conditions. Lanes: 1–wt (AB), 2–Tsp4b-deficient embryos injected with tsp4b-GFP mRNA, 3–Tsp4b-deficient embryos injected with SQAS-GFP mRNA, 4–Tsp4b-deficient embryos co-injected with tsp4b-FLAG and SQAS-GFP mRNAs. Pentameric Tsp4b-GFP (~663 kDa) and pentameric Tsp4b-FLAG (~535 kDa) bands (black arrow head). Monomeric SQAS-GFP (~132 kDa) (grey arrow head). The band corresponding to a 100 kDa size marker in all lanes of the blot reacted with anti-GFP is a background signal.

The following figure supplements are available for figure 6:

**Figure supplement 1**. *tsp4b-SQAS-gfp* mRNA is expressed similar to wild type *tsp4b-gfp* mRNA.

**Figure supplement 2**. The CQAC motif is essential for Tsp4b localization and function.

**Figure supplement 3**. Laminin and FAK localization are dependent on pentameric Tsp4b.

---

synthesized and secreted (*Figure 6—figure supplement 1*). However, at later stages while *tsp4b-gfp* localized to MTJs similar to wild-type Tsp4b, SQAS-gfp fluorescence was not localized and gradually disappeared (*Figure 6—figure supplement 2A–D*). SQAS-gfp also failed to localize to MTJs in Tsp4b-deficient embryos, in contrast to tsp4b-FLAG mRNA (*Figure 6—figure supplement 2E–H*). To confirm the role of CQAC motif in maintaining stable functional Tsp4b pentamers, we extracted whole embryo protein under non-reducing conditions, from 36 hpf Tsp4b-deficient embryos expressing Tsp4b-GFP, SQAS-GFP and co-expressing SQAS-GFP and Tsp4b-FLAG respectively. A western blot of the protein extract showed that Tsp4b-FLAG and Tsp-GFP predominantly existed as a pentamers, while SQAS-GFP was exclusively present as monomers, further confirming that the oligomerization function of Tsp4b mediated by CQAC motif of CCD is essential for localization of Tsp4b to the MTJ (*Figure 6C*).

To test requirements for oligomerization of Tsp4b in muscle attachment, ECM organization and Itg signaling we injected wild type embryos with the SQAS form of *tsp4b* mRNA. Injected larvae at 3 dpf showed significantly increased muscle detachment upon stimulation compared to controls. Furthermore, when co-injected with the *tsp4b* MO, the SQAS *tsp4b* mRNA failed to rescue muscle

detachment (*Figure 6B*), FAK activation or Lam localization at MTJs (*Figure 6—figure supplement 3*). These results suggest that in addition to a requirement for its own assembly and localization at MTJs, pentameric Tsp4b is a key scaffolding component that is required for MTJ ECM assembly and muscle-specific Itg signaling.

## Human TSP4 rescues muscle attachments in Tsp4b-deficient zebrafish

Studies of bovine, equine and human TSP4 have shown that it localizes to tendons (*Hauser et al., 1995*; *Chen et al., 2000*; *Södersten et al., 2006*). To determine if the functional roles of zebrafish Tsp4b are evolutionarily conserved, we injected recombinant human TSP4 (2 ng/embryo) into the interstitial space between myofibers of 60 hpf Tsp4b-deficient embryos. Exogenous TSP4 protein localized to MTJs adjacent to the site of injection and rescued muscle detachment in Tsp4b-deficient larvae when stimulated (22% detached [N = 136] compared to 52% detached [N = 96] in stimulated Tsp4b-deficient embryos) (*Figure 7A–I*). In addition, injection of exogenous TSP4 protein reduced the occurrence of muscle detachments by ~20% in wild-type embryos (N = 40 embryos) electrically stimulated with increasing voltages (*Figure 7J*). These results suggest that TSP4 and zebrafish Tsp4b are both structurally and functionally similar, and that TSP4 also serves as a scaffold for ECM assembly at MTJs.

## Discussion

Vertebrate tendons consist of a complex network of collagen-rich fibrils, which interact with other ECM proteins and membrane adhesion complexes on the surfaces of muscle cells to form attachments strong enough to bear contractile forces (*Ros et al., 1995*). How does such a complex network of ECM proteins assemble and maintain its organization? Muscles of the embryonic zebrafish undergo dynamic changes in ECM composition (*Henry et al., 2001*; *Crawford et al., 2003*; *Snow and Henry, 2009*; *Sen et al., 2011*), in which early high levels of Fn are replaced by Lams and Cols and later by an orthogonal array of collagen fibrils (*Kannus, 2000*; *Câmara-Pereira et al., 2009*; *Charvet et al., 2013*). These changes in ECM are critical for muscle attachment and maturation of MTJs. Here we show a novel role for pentameric Tsp4 as a key scaffolding protein that orchestrates the ECM organization of MTJs necessary for muscle attachments.

Because Tsps interact with other Itg ligands (e.g., Lam, Col, Fn) in vitro it has been debated as to whether Tsps play instructive or merely permissive roles in ECM organization and cell-ECM interactions (*Narouz-Ott et al., 2000*; *Södersten et al., 2006*; *Tan and Lawler, 2009*). In *Drosophila*, by virtue of its canonical RGD motif, Tsp plays a primary role as an Itg ligand in developing tendons. In contrast, our results suggest that zebrafish Tsp4 has a dual role, both binding Itgs through its non-canonical KGD domain and organizing the tendon ECM at MTJs to maintain muscle attachments. We propose a model in which pentameric Tsp4 in vivo functions as a scaffold to assemble other ECM components, and their interactions with Itgs and Dys complexes at MTJs (*Figure 8*). Our mutational studies of the CQAC motif support this hypothesis and show, for the first time in vivo, that pentamerization of Tsp4 is central to its scaffolding role, which organizes collagen fibrils in tendons in response to contractile force from muscles (*Hauser et al., 1995*; *Kjaer, 2004*; *Aparecida de Aro et al., 2012*).

Human TSP4, which forms pentamers (*Lawler et al., 1995*; *Frolova et al., 2012*), is also capable of rescuing Tsp4b-deficient zebrafish, demonstrating that the two are functionally interchangeable. Zebrafish Tsp4b shares the conserved C-terminal domain with human TSP4 and TSP5. The C-terminal region of mammalian Tsp5, which also functions as a pentamer, has been suggested to interact with multiple ECM proteins in cartilage (*Hecht et al., 1998*; *Chen et al., 2007*) and with collagen fibrils in tendons (*Chen et al., 2007*; *Södersten et al., 2007*; *Tan and Lawler, 2009*). We hypothesize that Tsp4 serves as a central organizing factor in MTJs by interacting with multiple ECM proteins, proteoglycans and receptors on muscle membranes. Other Tsps could play similar organizational roles as pentamers (subclass B) or trimers (subclass A) in contexts where they are required such as neurite outgrowth, synapse formation, wound healing and tumorigenesis (*Arber and Caroni, 1995*; *Kyriakides et al., 1998*, *1999*; *Sid et al., 2004*; *Christopherson et al., 2005*).

Analysis of Tsp4b expression in zebrafish reveals surprising differences in muscle-tendon cell interactions during MTJ development in different muscle groups. Axial muscles attach directly to the basement membrane at somite boundaries in the early embryo and become contractile and functional at least 12 hr before tenocytes are detected at MTJs (*Devoto et al., 1996*; *Barresi et al., 2001*; *Henry et al., 2005*; *Snow and Henry, 2009*). Our results suggest that after axial muscle progenitors attach, they secrete their own Tsp4b, which they downregulate upon differentiation, while later Tsp4b is secreted

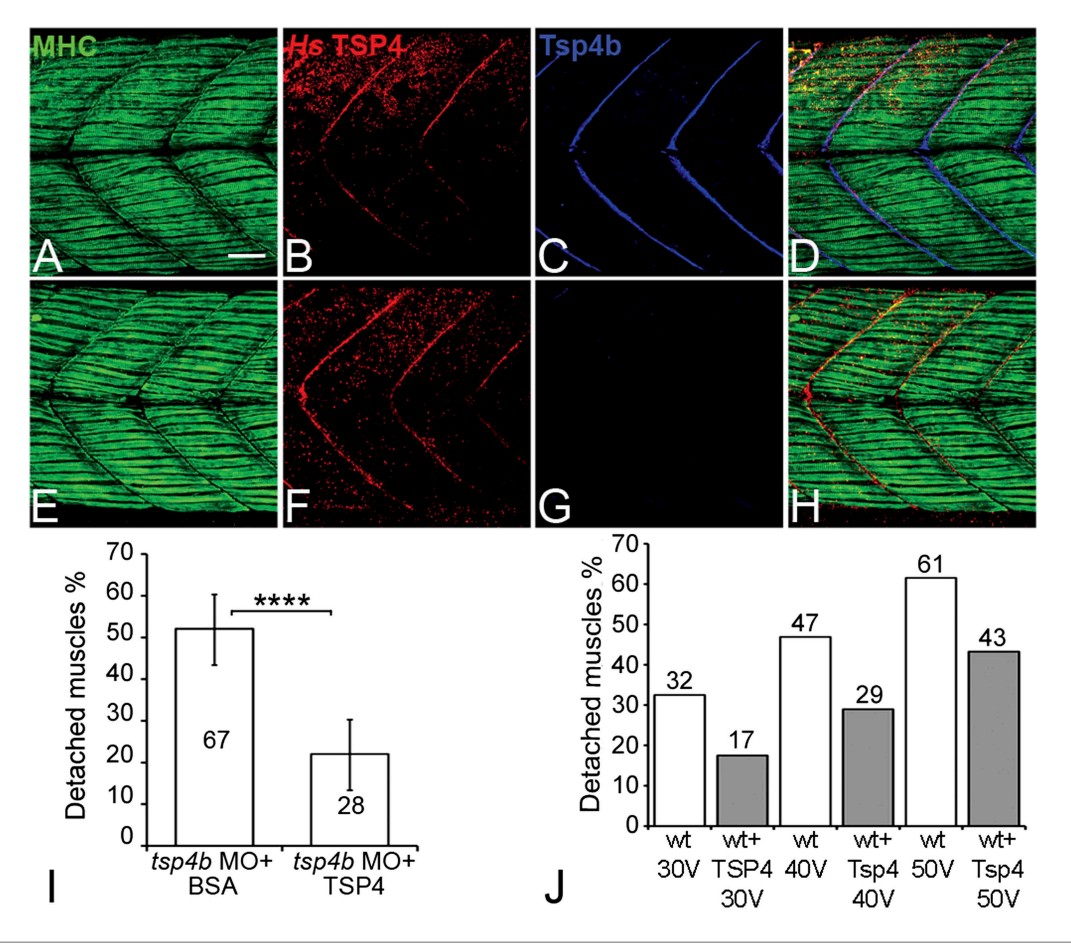

**Figure 7**. Conserved functions for TSP4 in maintenance of MTJs. (**A–H**) Lateral views of trunk muscles in 72 hpf embryos stained with anti-MHC (green), anti-human TSP4 (red) and anti-Tsp4b (blue) antibodies. (**A–D**) Injected recombinant human TSP4 co-localizes with zebrafish Tsp4b at muscle attachments in wild type embryos. (**E–H**) Injected TSP4 localizes to somite boundaries in Tsp4b-deficient embryos. (**I**) Injected TSP4 rescues muscle attachments in Tsp4b-deficient embryos upon stimulation (N = 96 embryos) (Chi squared test, p value<0.001). (**J**) Histogram showing percentage of embryos with detached muscles in 60 hpf wt+BSA (white columns) and wt+TSP4 (shaded columns) embryos, stimulated at 30 V, 40 V and 50 V, respectively. N = 40 embryos for each sample. (Scale bars = 30 microns).

and maintained by tenocytes. Tenocytes in the zebrafish trunk likely originate from the syndetome compartment in each somite, similar to their counterparts in the chick, and migrate to sites of muscle attachment (**Brent et al., 2003**; **Shukunami et al., 2006**; **Chen and Galloway, 2014**). In contrast, we observe Tsp4b expression in cranial tenocytes at MTJs but not in cranial muscles. This difference in the timing and sources of Tsp4b between cranial and axial muscles suggests that there are distinct types of interactions between myoblasts and tenocytes involved in forming different types of MTJs. Our results suggest that Tsp4b is required for both initial attachments of some early muscles, which correlates with defects in early ECM, as well as more broadly in the maintenance of all types of attachments.

$Tsp4^{-/-}$ mice show progressive weight loss, atrophy of muscle mass and disrupted packing of collagen fibrils, but they are viable and fertile (**Frolova et al., 2010**, **2014**). Similarly, Tsp4b-deficient zebrafish appear largely unaffected until their muscles are subjected to multiple contractions. Stress-induced expression appears to be a common feature of many Tsps. For example, TSP4 (and other Tsps) is upregulated in response to exercise, myocardial infarction and in several types of muscular dystrophy (**Chen et al., 2000**; **Timmons et al., 2005**; **Frolova et al., 2012**). Tsp5 expression is elevated in response to biomechanical stress (**Hecht et al., 1998**). Subclass A Tsps (Tsp 1–2) are also elevated in response to injury or inflammation (**Bornstein et al., 2004**). Elevated Tsp expression in all these scenarios could reflect an attempt by cells at ECM repair but may also be detrimental by causing tissue fibrosis. Consistent with this

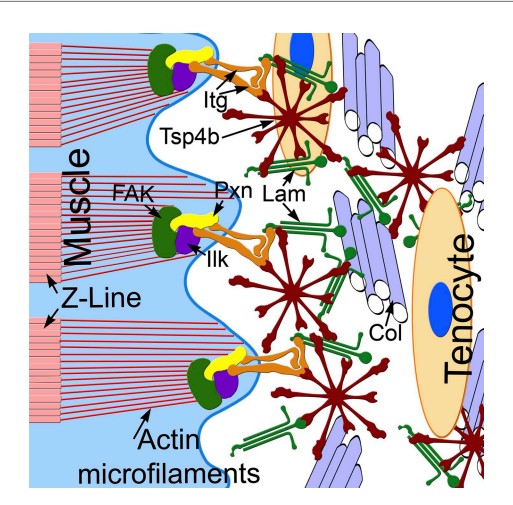

**Figure 8**. Tsp4b establishes and maintains MTJ ECM organization. A model for ECM assembly at an MTJ. Pentameric assemblies of Tsp4b (red) associate with Lam (green), Col fibrils and other ECM components. Tsp4b and Lam bind Itgs (orange) on the muscle cell surface, activating FAK (green) and recruiting Pxn (yellow) and Ilk (purple) to promote muscle specific Itg signaling and stabilize myofiber attachment.

idea, in Tsp2⁻/⁻ mutant mice, wound healing progresses rapidly without scarring, but with abnormal Col fibril structure (**Kyriakides et al., 1998**; **Bornstein et al., 2004**). $Tsp4^{-/-}$ mice show elevated levels of Col (Col I, II, III, V) in the heart. This elevated accumulation of Col is reversed with application of TSP4 in an in vitro cell culture system (**Frolova et al., 2012**). This result is similar to our in vivo study, where injected TSP4 rescues the Tsp4b-deficient phenotype, suggesting that the scaffolding function of Tsp4 is evolutionarily conserved and functions as a modulator of ECM organization in multiple tissue types. Recent work on the functions of Tsp4 in cardiac ECM has shown interesting intracellular functions, where Tsp4 localized to the endoplasmic reticulum of cardiac fibroblasts promotes nuclear shuttling of activating transcription factor 6 alpha (**Lynch et al., 2012**). While this intracellular role of Tsp4 cannot be ruled out in MTJ organization, the ability of exogenous TSP4 protein injected extracellularly to rescue a Tsp4b-deficient phenotype suggests that the primary function of Tsp4 is in ECM organization at MTJs.

Muscular dystrophies are characterized by loss of muscle activity due to defects in interactions between muscles and the myomatrix ECM at MTJs. A mouse model for congenital muscular dystrophy caused by Lama2 deficiency can be treated effectively by injecting recombinant Laminin1-1-1 trimer (**Rooney et al., 2012**). Hence, the strategy of treating such 'myotendinopathies' by injecting recombinant ECM proteins is a viable option. In addition, sports- and age-related tendon injuries are often difficult to treat and require prolonged recovery periods. Our work provides a potential new therapeutic strategy using TSP4 to promote efficient tendon and MTJ repair and correcting defects in the ECM in a whole range of myotendinopathies.

## Materials and methods

### Fish lines

Wild type embryos of the *tupfel long fin* (*TL*) line were used for all experiments with the exception of *lost contact* (*loc*) mutants, which disrupt *integrin-linked kinase* (*ilk*) (**Postel et al., 2008**), tg(itga5:tagrfp) (**Lackner et al., 2013**) and *tg(bactin:pxn-egfp)* transgenics (**Goody et al., 2010**). All embryos were raised in standard embryo medium at 28.5°C (**Westerfield, 2007**) and staged as described previously (**Kimmel et al., 1995**). Craniofacial muscles of zebrafish embryos were labeled as described previously (**Schilling and Kimmel, 1997**). Adult fish and embryos were collected and processed in accordance with the rules and protocols approved by UCI-IACUC.

### Morpholinos and mRNA injections

An antisense morpholino (MO) oligonucleotide—CGGCCATCCTTCAATCACAACCTTC—was designed against the translation start site of *tsp4b* by Gene Tools, LLC. To reduce MO-induced cell death a *p53*-MO (AGAATTGATTTTGCCGACCTCCTCT) was co-injected at 0.35 ng/embryo. All primers used are listed in **Supplementary file 1**. The full-length *tsp4b* mRNA used in rescue experiments was synthesized from a full-length *tsp4b* cDNA clone (Cat No. MDR1734-202795262; Fisher Scientific, Pittsburgh, PA, USA) in *pBS SK+* vector, using T3 RNA polymerase (mMessage mMachine T3 transcription kit AM1348, Life Technologies, Grand Island, NY, USA) and tailed (Poly(A) tailing kit, AM1350 Life Technologies, Grand Island, NY, USA). The *pCS2-itga6-GFP* clone was prepared using CloneEZ (Cat No: L00339; GenScript USA Inc., Piscataway, NJ, USA). The cloning followed a two-step protocol. In the first step, *pCS2* vector was cut at ClaI site and *itga6* amplicon synthesized with complementary overhangs was cloned. The

following primers were used for amplifying *itga6*; FP: gttcttttgcaggatcccat**atcgat**ATGGAGCTCTTTG AGAAAGCGC (lower case—*pCS2* sequence, bold—ClaI site, upper case—*itga6* sequence) RP: GAGAGGCCTTGAATTCGA**atcgat**tgagtagctctcgttctcatcccac (lower case—*pCS2* sequence, bold—ClaI site, upper case—*itga6* sequence). The *gfp* amplicon synthesized with complementary overhangs was cloned into a *pCS2-itga6* clone at a BstBI site. The following primers were used for amplifying *gfp* from *pEGFP-C1* vector; FP: CGAGAGCTACTCAATCGA**TTTA**<u>ATGGTGAGCAAGGGCGAGG</u> (upper case— *tsp4b* sequence, bold—bridge to maintain translation frame, upper case under lined—*gfp* sequence); RP: ccttgaattcgaatcgatg**gaattc**CTATAGGGCTGCAGAATCTAGAGGCTCG (lower case—*pCS2* sequence, bold—EcoRI site, upper case—*gfp* sequence).

## Knockdown and rescue

*tsp4b* antisense morpholino (MO) oligonucleotide was injected at either 0.16 or 0.32 ng/embryo; the former reduced and the latter eliminated Tsp4b protein as assayed immunohistochemically. For rescue experiments zebrafish *tsp4b* mRNA was synthesized from a full-length cDNA clone. *KGE* and *SQAS* mutant mRNAs were synthesized from the full-length *tsp4b* cDNA clone using modified site-directed mutagenesis protocol (Cat No: 200555 Quikchange II-E site-directed mutagenesis kit Agilent Technologies, Santa Clara, CA, USA) with the following primers: *KGE* forward-GGGAAGGGTGA**G**GCATGTG; *KGE* reverse-CACATGC**C**TCACCCTTCCC; *SQAS* forward-CATCTCTGAG**A**GCCAGGCC**A**GCGGGCTGAG; *SQAS* reverse- CTCAGCCCGC**T**GGCCTGGC**T**CTCAGAGATG. *KGE* mutated site (in bold underlined)— C1482G Asp>Glu; *SQAS* mutated sites (in bold underlined)—T781A and T790A, Cys>Ser. All RNAs were injected at 100 pg/embryo. The *gfp* and *FLAG* tagged constructs were constructed using Gibson Cloning on *pCS2* vector backbones (*Gibson et al., 2009*). The following primers were used to amplify the *pCS2* vector backbone, *tsp4b* mRNA, *gfp* and *pCS2-3XFLAG* vector backbone. *pCS2* vector for *tsp4b-gfp* fusion: FP—<u>GAGCCTCTAGATTCTGCAGCCCTATAG</u>gaattcaaggcctctc gagcctctag (upper case under lined is *gfp* sequence, lower case is *pCS2* vector sequence); RP— GAGGAGATGCATTGTGCCGGCCATgaatcgatgggatcctgcaaaaagaacaag (upper case is *tsp4b* sequence, lower case is *pCS2* vector sequence). *tsp4b* for *gfp* fusion: FP—gttcttttgcaggatcccatcgattcATGGC CGGCACAATGCATCTCC (upper case is *tsp4b* sequence, lower case is *pCS2* vector sequence); RP—<u>GCTCCTCGCCCTTGCTCACCAT</u>CAAGGGGTCCATGCCATGTTGTGTACTG (upper case under lined is *gfp* sequence, upper case is *tsp4b* sequence). The same set of primers was used to amplify the *SQAS* mutant form of cDNA from the *pCS2* clone. *gfp* for *tsp4b* fusion: FP—GTACACAACATGGCAT GGACCCCTTG<u>ATGGTGAGCAAGGGCGAGGAGC</u> (upper case is *tsp4b* sequence, upper case under-lined is *gfp* sequence); RP—ctagaggctcgagaggccttgaattc<u>CTATAGGGCTGCAGAATCTAGAGGCTC</u> (upper case under lined is *gfp* sequence, lower case is *pCS2* vector sequence).

 *tsp4b* for *FLAG* fusion: FP—caagctacttgttcttttgcaggatcATGGCCGGCACAATGCATCTCCTC (upper case is *tsp4b* sequence, lower case is *pCS2-3XFLAG* vector sequence); RP—cgtcatggtctttgtagt ccatgtcCAAGGGGTCCATGCCATGTTG (upper case is *tsp4b* sequence, lower case is *pCS2-3XFLAG* vector sequence). *pCS2-3XFLAG* for *tsp4b* fusion: FP—CAACATGGCATGGACCCCTTGgacatggacta caaagaccatgacg (upper case is *tsp4b* sequence, lower case is *pCS2-3XFLAG* vector sequence); RP— GAGGAGATGCATTGTGCCGGCCATgatcctgcaaaaagaacaagtagcttg (upper case is *tsp4b* sequence, lower case is *pCS2-3XFLAG* vector sequence).

 The synthesized mRNA was purified and concentrated using RNA clean and concentrator-5 kit (Cat No: R1015; Zymo Research, Irvine, CA, USA). Recombinant human TSP4 (TSP4) protein solution was injected at 1 mg/ml, using a glass microelectrode, into the interstitial space between myofibers of anesthetized 48 hpf embryos. Embryos were allowed to recover for 12 hr in normal embryo medium before fixation for further analysis.

## Muscle stimulation

To assess muscle attachment strength we designed an electrical stimulation protocol to induce muscle contraction. Teflon-insulated platinum microelectrodes (D10PW; Plastics One Inc., USA), with the insulation stripped from the ends, were connected to a Grass SD-5 stimulator (Grass Instruments, Warwick, RI, USA) with which we could vary pulse strength (volts), duration, frequency (λ), and delay. Electrodes were positioned at the anterior and posterior ends of individual wild type or Tsp4b-deficient embryos and stimulated at varying strengths—0 V, 20 V, 30 V, 40 V, 50 V—at duration = 8 ms, λ = 4 pulses/s, and delay = 6 ms. Embryos were anaesthetized with Tricaine (ethyl 3-aminobenzoate methanesulfonate; Cat No. A5040, Sigma-Aldrich, Milwaukee, WI, USA) and placed on a silicone plate with embryo medium and stimulated

for 2.5 min before transfer into fixative. 30 V was found to be most effective in causing muscle detachment but allowing embryos to remain otherwise healthy with this protocol (10% of wild type embryos show any detached muscles, while 60% of Tsp4b-deficient embryos show widespread detachment in somites). To determine the dose–response, *tsp4b*-MO embryos injected with different amounts were subjected to muscle stimulation–28% (N = 7/25) injected with 0.16 ng/embryo of *tsp4b*-MO showed detachment, 63% (N = 40/64) injected with 0.32 ng/embryo showed detachment.

## Whole mount in situ hybridization and immunohistochemistry

Zebrafish *tsp4b* (NM_173226) expression was visualized using an antisense RNA probe synthesized against a 626 bp (10–636 bp) region of *tsp4b* cDNA using T7 RNA polymerase (Cat No: 10881767001, Roche Diagnostic Corporation Indianapolis, IN, USA). Zebrafish *tnmd* (NM_001114413) expression was visualized using an antisense RNA probe synthesized to recognize a 602 bp (109–710 bp) region of *tnmd* cDNA using T7 RNA polymerase. Whole mount in situ hybridization was performed according to standard protocol and developed using NBT/BCIP (NBT-11383213001, BCIP- 11383221001, Roche Diagnostics Corporation, Indianapolis, IN, USA) or Fast Red (Cat No: 11496549001, Roche Diagnostics Corporation, Indianapolis, IN, USA) (*Thisse and Thisse, 2008*). A zebrafish-specific Tsp4b antibody was generated against 618 bp (91–708 bp) of the unique N-terminal region. This was cloned into pGEX-4T-2 expression vector, expressed as a GST tagged peptide, purified as per standard protocol and this fusion protein (206 aa) was used to raise antibodies in rabbits at Thermo Fischer/Open Biosystems, Rockford, IL, USA (*Ring et al., 2002*). Wild-type embryos stained with pre-immune serum from host showed absence of specific signal. Antibody specificity was verified by staining wild-type and Tsp4b deficient embryos with anti-Tsp4b. All embryos used for immunofluorescence experiments were fixed in 95% methanol and 5% glacial acetic acid for 4–6 hr at −20°C. They were rehydrated in Phosphate Buffered Saline (PBS) with 2% dimethyl sulfoxide (DMSO), 0.5% Triton (PBDT) and permeabilized with cold acetone for 10–15 min at −20°C. Following permeabilization, a standard antibody staining protocol with PBDT was used. Primary antibodies and concentrations used: rabbit anti-Tsp4b (1:500), rabbit monoclonal anti-human TSP4 (1:300; Abcam, Cambridge, MA, USA), mouse anti-myosin heavy chain (MHC) (A1025; Developmental Studies Hybridoma Bank, Iowa City, IA, USA - 1:250), rabbit anti-pan laminin (*Peterson and Henry, 2010*) (RB-082-A0; 1:250; Thermo Scientific Inc., Waltham, MA, USA), rabbit anti-FAK [pY861(*Henry et al., 2001*) [44-626G], 1:250; Life Technologies, Grand Island, NY, USA], and chicken anti-GFP (ab13970; 1:1000; Abcam, Cambridge, MA, USA). DiAmino PhenylIndole (DAPI) (D1306; 1:1000; Life Technologies, Grand Island, NY, USA) was used to mark cell nuclei. Preabsorbed secondary antibodies were all obtained from Jackson Immunoresearch, West Grove, PA, USA and used for indirect immunofluorescence at 1:1000, including: Alexa Fluor 488 conjugated donkey anti-mouse IgG (715-546-150), DyLight 549 conjugated donkey anti-rabbit IgG (711-506-152), Alexa Fluor 488 conjugated donkey anti-chicken IgY (703-486-155), Alexa Fluor 488 conjugated donkey anti-mouse IgG (715-546-150). After staining, embryos were mounted in 1% low melt agarose in PBS and imaged. The quantity of embryos chosen was dependent on the variance observed in the experiment. Experiments that involved stimulation of muscles after various treatments were performed on a larger data set to account for variability in the quantity and quality of treatments. Embryos that exhibited gross morphological defects associated with RNA toxicity or morpholino toxicity or other idiopathic growth defects were excluded from the analysis. In order to avoid background effects of adult parents, embryos from different tanks with adults of different age groups were used to collect the embryos for the analysis.

## Western blot

Tsp4b forms pentamers through inter-chain interactions at the CCD and formation of inter-chain disulfide bonds with cysteine residues of the CQAC motif. In order to visualize the pentameric form on the blot it was necessary to perform the protein extraction and western blot protocol in a non-reducing condition (*Narouz-Ott et al., 2000*). Whole protein extract was prepared from 36 hpf embryos, de-yolked in Ringer's solution by rapid flushing using a pipette tip. The de-yolked embryos were pelleted at 1500 RPM to remove the yolk granules in the supernatant. After three washes in Ringer's solution, a modified radio immunoprecipitation assay (RIPA) buffer was added (2 µl per embryo). RIPA composition: 0.05 M Tris pH 7.4, 0.15 M NaCl, 1% Triton X100, 1.25 mM $CaCl_2$, 0.9 mM $MgCl_2$, 20 µM $MnCl_2$, 0.2 mM $ZnCl_2$ and protease inhibitors (PMSF and Leupeptin). The tissue was homogenized using a pestle and DNase added to the homogenate and incubated on ice for 30 min. A 3–10% gradient gel was cast using a gradient former. A non-reducing protein loading buffer was added to the protein extract and 20 µl of this protein extract was loaded on to the gel. A prestained PageRuler protein ladder (Cat No. 26616; Thermo

Scientific Inc., Waltham, MA, USA) was used as a molecular weight reference. Western analysis was performed on the blot using mouse monoclonal anti-FLAG antibody (Clone M2; Cat No. F1804; Sigma-Aldrich, Milwaukee, WI, USA) and mouse monoclonal anti-GFP antibody (Clone JL-8; Cat No. 632381; Clontech) at 1:1000 dilution and secondary antibody anti-mouse-HRP (715-035-150; Jackson Immunoresearch, West Grove, PA, USA) at 1:10,000 dilution. Signal development was performed using a chemiluminescent assay with Luminol (Cat No. 123072, Sigma-Aldrich, Milwaukee, WI, USA) and Coumaric acid (Cat No. C9008, Sigma-Aldrich, Milwaukee, WI, USA).

### qRT-PCR

Whole embryo RNA was extracted from wild-type and Tsp4b-deficient embryos were collected at 60 hpf according to standard protocols using Trizol (15596-018; Life Technologies, Grand Island, NY, USA). The experiment was designed in accordance with guidelines to conduct and report quantitative PCR (*Bustin et al., 2009*). The RNA quality was assessed on an Agilent bioanalyzer 2100 and a RIN value >9.0 was obtained for the samples. RNA concentration was normalized between samples and used as a template for cDNA synthesis. cDNA was synthesized with oligo dT primers using the standard protocol of ProtoScript M-MuLV First Strand cDNA Synthesis Kit (#E6300S; New England BioLabs Inc., Ipswich, MA, USA). The synthesized cDNA was diluted to 1:20 and used as a template for qRT-PCR using the protocol for LightCycler 480 SYBR Green I Master kit (04707516001; Roche). The reaction was run on LightCycler 480 II Real time-PCR Instrument (Roche Diagnostics Corporation, Indianapolis, IN, USA) and analyzed using LightCycler 480 Software release 1.5.0 SP3. The histogram was plotted on Microsoft Office–Excel.

### Human TSP4 rescue

Lyophilized recombinant human TSP4 without the signal peptide (Ala22-Asn961, accession # P35443) tagged with a C-terminal His tag was derived from Chinese Hamster Ovary (CHO) cell line (2390-TH-050 Lot #MLO0612091; RD systems Inc., Minneapolis, MN, USA) It was reconstituted in a tris buffer (*Narouz-Ott et al., 2000*) (15 mM Tris pH 7.4, 75 mM NaCl, 1 mM $ZnCl_2$, 1 mM $CaCl_2$) at a concentration of 1 mg/ml as per the manufacturer's instructions. Control experiments were performed by injecting BSA solution in the same buffer at a 1 mg/ml concentration. Wild-type and Tsp4b-deficient 36 hpf embryos were anesthetized in tricaine and embedded in 2% low melting agarose (prepared using embryo medium). Using a microelectrode, 2 ng of human TSP4 protein is injected into the extracellular space between muscle fibers. Embryos were removed from the agarose and transferred to embryo medium and allowed to recover for 12 hr in 28°C. Approximately half of wild-type and Tsp4b-deficient embryos injected with the human TSP4 protein were subjected to mild electrical stimulation and fixed for staining.

### Transmission electron microscopy (TEM)

Unstained 72 hpf wild-type and Tsp4b-deficient embryos (both before and after stimulation) were fixed in a 4% PFA (Cat # 15710; Electron Microscopy Sciences [EMS], Hatfield, PA, USA) and 2% Glutaraldehyde (Cat # 16020; EMS, Hatfield, PA, USA) cocktail for 6 hr at 4°C. Embryos were washed in PBS+0.1% Triton and embedded in 5% low melt agarose (Cat #20-104; Apex Chemicals). Vibratome sections (Leica VT1000S) were cut in cold PBS. Lateral sections (75 µm) were cut at 0.75 mm/s and a vibration frequency of 70 Hz. The sections were washed in 0.15 M Sodium Cacodylate (Cat # 12300; EMS, Hatfield, PA, USA), stained sequentially with 1% Osmium tetroxide (Cat No. RT 19100, EMS, Hatfield, PA, USA) and 1% Uranyl acetate (Cat No. RT 22400-1, EMS, Hatfield, PA, USA), and were embedded in Durcupan for ultrathin sections. The Imaging was performed on a JEOL 1200EX TEM (NCMIR, UCSD).

### Microscopy and image analysis

Whole mount in situ stained embryos were photographed using a Zeiss Axioplan-2 microscope using Volocity image acquisition software (Improvision, Perkin Elmer Inc.). Embryos processed for fluorescent immunohistochemistry were imaged using either: (1) an Olympus Fluoview FV100 confocal system with an IX81 inverted microscope using Plan-Apo 20X/0.75 NA and Plan-Apo 40X/1.3 NA (oil) objectives, respectively, or (2) a Zeiss LSM780 confocal system with an Observer Z1 inverted microscope and a Plan-Apo 20X/0.8 NA objective. Confocal stacks were analyzed using ImageJ software (*Hartig, 2013*).

### Sequence analysis

Multiple sequence alignment of protein sequences was performed using ClustalW2 (http://www.ebi.ac.uk/Tools/msa/clustalw2/). Phylogenetic tree of Tsp genes was constructed using MegAlign on DNASTAR Lasergene suite.

## Statistical analysis

Quantification of data from muscle stimulation studies was performed with Microsoft Excel (Office 2007). Statistical significance was calculated using a Chi-Squared test. Standard error calculated from the data set and plotted on the histogram.

## Quantification of fluorescence intensity (FI)

In order to quantify protein localization, FI was measured within the area encompassing the somite boundary (Area-SB) in the embryo and compared with a similar area away from the boundary (Area S). Using the freehand drawing tool in ImageJ, a region encompassing the dorsal or ventral half of the SB was drawn, or the mid-section of S. Mean FI was measured in arbitrary units (A.U.) using the Measure function under Analyze tab in ImageJ. Since, the area under each SB is different, the mean FI for each SB is calculated to an area that is normalized with the area of S. The FI of localized protein = FI of Area SB − FI of Area S. FI was calculated for three somite boundaries per embryo, averaged and placed on a scatter plot.

Statistical significance of the distribution of FI between wild type and Tsp4b-deficient embryos was calculated by paired Student t-test with unequal variance. In experiments, where statistical significance is calculated using *t* test, ANOVA single factor analysis was performed with posthoc multiple comparisons using Tukey method. ANOVA analysis was performed using 'Daniel's XL Toolbox version 6.22' add-in for Microsoft Excel. In order to prevent bias during data collection, embryos were from different samples were first observed under the dissecting microscope and enumerated before establishing the sample identity.

## Quantification of MTJ basement membrane width

The images chosen for quantification were of MTJs in the mid-section of a dorsal or ventral SB, at least a couple of cell diameters lateral to the edges of the notochord. The TEM images have a scale bar that is set for 0.5 microns. Using Set scale function under Analyze tab in ImageJ, the number of pixels along the length of the scale bar was recorded for each image. Using the line draw function, lines were drawn spanning the edges of the basement membranes at the three widest regions in the image. The number of pixels spanning the length of these lines were recorded and averaged for each image. The width of the basement membrane was calculated by: (total pixels along an average width of an MTJ − 0.5 micron)/(total pixels along the length of the scale bar). In most of the stimulated Tsp4b-deficient embryos, the MTJ gaps were very wide and not captured in a single image and hence not quantified.

## Acknowledgements

We thank AThor and E Bushong from NCMIR at UCSD for assistance in TEM sample processing and imaging; R Meyer for assistance in constructing the electrical stimulation apparatus; M Waterman for advice on designing western blot experiments; C Rackauckas for advice on statistical analysis; T Volk, S Nair, A Muto, S Piloto, K Radtke, C Alexander and members of the lab for comments on the manuscript; T Zhang and I Gehring for fish care. This work was supported by NIH grants R01 DE013828 and R21 AR062792 to TFS.

## Additional information

### Competing interests

TFS: TFS has filed a provisional patent 'Thrombospondin proteins and methods of using for treating tendons and ligaments' with US application serial no. 61/835,325. The other authors declare that no competing interests exist.

### Funding

| Funder | Grant reference number | Author |
| --- | --- | --- |
| National Institutes of Health (NIH) | R21 AR62792, R01 DE13828 | Thomas F Schilling |

The funder had no role in study design, data collection and interpretation, or the decision to submit the work for publication.

## Author contributions

AS, Contributed towards intellectual discussions leading to design of experiments, Performing experiments, Data acquisition, Interpretation of data, Revising the manuscript; TFS, Contributed towards intellectual discussions leading to design of experiments, Interpretation of data, Performing cell transplantation experiments, Revising the manuscript

## Author ORCIDs

Arul Subramanian, http://orcid.org/0000-0001-8455-6804

## Ethics

Animal experimentation: This study was performed in accordance with rules and protocols approved by University of California, Irvine- Institutional Animal Care and Use Committee (UCI-IACUC) (Protocol # 2000-2149-4). Juveniles and adult fish were euthanized with Ethyl 3-aminobenzoate methanesulfonate (Tricaine). Embryos were anesthetized with Tricaine before stimulation assays.

## Additional files

### Supplementary file

• Supplementary file 1. List of primer sequences used for qRT-PCR, cloning of mRNA and probe synthesis.

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
