## [Decision Letter]

Thank you for sending y“our work entitled “Thrombospondin-4 controls matrix assembly during development and repair of vertebrate myotendinous junctions.” for consideration at *eLife*. Your article has been favorably evaluated by a Senior editor and 3 reviewers, one of whom is a member of our Board of Reviewing Editors.

The Reviewing editor and the other reviewers discussed their comments before we reached this decision, and the Reviewing editor has assembled the following comments to help you prepare a revised submission.

This is an interesting study providing convincing evidence for a role of Thrombospondin 4b in muscle attachment in the zebrafish embryo. As in Drosophila, Thrombospondin is required for stable muscle attachments in vertebrates. Human *tsp4b* RNA rescues zebrafish *tsp4b* morphants suggesting conservation of function. Elegant mosaic analysis shows regional rescue of muscle around Tsp4b-expressing cells. Also demonstrated clearly are effects through Tsbp4 on other integrin mediated events both inside and the cell (FAK phosphorylation and paxillin localisation) and outside the cell, laminin organisation. The manuscript has the potential to be stunning and impactful, adding to our understanding of muscle and tendon development and function in the zebrafish model, and has relevance and implications for the study of human muscle/tendon disease and injury.

In general, the work is careful and thorough, the figures are clear, and the conclusions are well supported by the data shown. However, the citation of others' work could be improved and the statistical analyses need attention before publication.

Specific comments:

1) The authors assume that Tsp4b only interacts with laminin and/or collagen binding integrins. Why is this assumption made given interactions between thrombospondins and Fibronectin? The tools exist for the authors to investigate which Integrins interact with Tsp4b. Determining mechanisms would expand the scope. If authors have a sound argument for not investigating Fn this needs to be presented in a coherent fashion.

2) There are two different phenotypes shown. One phenotype is muscle detachment. The conclusion is that “Our results suggest that Tsp4b is required to strengthen and maintain both types of attachments, but not for tenocyte formation or initial MTJ formation”. The other phenotype contradicts the above statement - there are discontinuous MTJs early in muscle development (20-24hpf). Tsp4b could play roles in both events - development of MTJ and strengthening/maintenance of attachment - but the data presented need to be clearly interpreted.

2a) Discontinuous MTJs: The authors show that Paxillin does not localize to MTJs at 20hpf, there is patchy laminin expression, and Fak phosphorylation is disrupted. The Time Axis needs to be considered. Are the initial somite boundaries forming but not being maintained? Are they not forming? Why are some muscle cells longer than others? And why is this phenotype not reflected in images shown in the beginning? This is a fundamental issue as the phenotype and conclusions drawn change throughout the manuscript. Does a lower dose lead to muscle detachment upon electrostimulation and a higher dose lead to developmental defects?

3) Lack of integration with the literature: “While the organization of the ECM at myotendinous junctions has been descripted, the developmental processes underlying its establishment and maintenance are poorly studied.” If the purpose of the manuscript is to further understand ECM establishment and maintenance at the zebrafish MTJ a logical approach would be to cogently describe what is known in the fish model. This would include that somites form, what is known about integrin signaling and matrix changes as MTJs form, what ECM proteins are known to be at the fish MTJ, phenotypes when these proteins are disrupted, and pathways that regulate their expression. While the authors are completely correct that MTJ development is not well understood, they should at least establish what is known in the field and how their results contribute. This is also true in the Discussion - multiple labs investigate ECM at these boundaries and have clearly laid out a time course and some cellular and molecular mechanisms of Fn and laminin deposition at these boundaries. Where does tsp4b fit into these pathways?

4) The conclusion that pentamerisation of Tsp4 is required for muscle-specific Itg signalling is not well supported (see comments for Figure 7 below) and this should either be toned down or strengthened by further experiment.

5) Can the authors explain their choice of mutation from KGD to KGE? Given that KGD appears to substitute for RGD, it is perhaps surprising that KGE fails to substitute for KGD, given that D and E amino acids are closely related.

6) Figures

Figure 1—figure supplement 2:The description here refers to “isolated cells”, but they look like groups of cells in the figure. Please clarify.

Figure 5 and Figure 6: It would be helpful to have the fluorescent images of the MO+RNA-injected embryos, in addition to the controls and the MO only-injected embryos (panels 5I and 6I) and to support the data in panel 5J.

Figure 7 legend: The legend for panel B should indicate that these embryos have undergone stimulation, as stated in the text.

Figure 7 is not convincing. Data for MO-only injected embryos should be shown (as for Figure 5), and rescue should be measured against these embryos rather than uninjected controls. Significance values between the experimental and control samples on this graph are weak, indicating that, in fact, the SQAS RNA can effect a pretty reasonable rescue of both FAK and Laminin levels.

Figure 8: Are these panels showing stimulated muscles, as indicated in the legend, and if so, using what voltage?

---

## [Author Response]

*1) The authors assume that Tsp4b only interacts with laminin and/or collagen binding integrins. Why is this assumption made given interactions between thrombospondins and Fibronectin? The tools exist for the authors to investigate which Integrins interact with Tsp4b. Determining mechanisms would expand the scope. If authors have a sound argument for not investigating Fn this needs to be presented in a coherent fashion*.

We do not intend to state that Tsp4b interacts with only laminin or collagen. Tsp interactions with different matrix proteins have been well studied. We have tried to examine Fn localization in Tsp4b deficient zebrafish embryos, using different fixation conditions (4% PFA, acid-methanol, TCA) but we have had trouble in getting available antibodies to work. We tried anti-Fibronectin (MP Biomedicals LLC, Cat No. 55066), anti-Fibronectin H300 (Santa Cruz biotechnology Cat No. SC-9068), and anti-Fibronectin (Sigma Aldrich Cat No. F3648), but all three failed to show any specific signal at the somite boundaries in wt embryos (Garavito-Aguilar et al., 2010). Previous studies have shown that rat Tsp4 interacts with laminin (Lam), fibronectin (Fn), collagens (Col), and matrillin in vitro (64). Zebrafish somite boundaries contain Lam, Fn and Cols, which become highly localized to the boundaries in larvae older than 3 dpf.

Availability of suitable antibodies has also hampered our ability to study integrins (Itgs) in zebrafish. Among three antibodies against focal adhesion kinase (FAK) – Anti-FAK (pY397) (Cat No: 44-625G), Anti-FAK (pY576) (Cat No: 44-652G), Anti-FAK (pY861) (Cat No: 44-626G) from Invitrogen - Anti-FAK (pY861) showed specific localization to basal ends of muscle fibers at MTJs. This demonstrated that activated Itg signaling requires Tsp4b signaling (Figure 4).

In our attempt to identify Itg receptors for Tsp4b we have now examined expression and localization of two likely candidates, Itga5 and Itga6, both expressed in embryonic mesoderm (Goody et al., 2012; [58]). We obtained *Tg(Itga5-rfp)* (gift from S Holley) and with live confocal imaging found that Itga5 localizes to the basal ends of muscles as they attach. *itga5-rfp* expression was modestly reduced at the membrane of Tsp4b-deficient embryos, consistent with a function as a Tsp4b receptor (Figure 4—figure supplement 2). We also prepared a fusion construct of *itga6-gfp* from a full length cDNA clone of *itga6* and injected in vitro-synthesized mRNA into wt embryos. We found that itga6-GFP localized along the entire length of muscle fibers, not just at their ends as we might imagine for a Tsp4b receptor. Itga6-GFP localization was also slightly reduced in Tsp4b-deficient embryos (Figure 4—figure supplement 2). These results have been added to the manuscript. From these results, we can postulate that similar to its interactions with Lam, itga5 is a good candidate for a Tsp4b receptor.

*2) There are two different phenotypes shown. One phenotype is muscle detachment. The conclusion is that “Our results suggest that Tsp4b is required to strengthen and maintain both types of attachments, but not for tenocyte formation or initial MTJ formation”. The other phenotype contradicts the above statement - there are discontinuous MTJs early in muscle development (20-24hpf). Tsp4b could play roles in both events - development of MTJ and strengthening/maintenance of attachment - but the data presented need to be clearly interpreted*.

Previous studies on early development of somites have shown that the ECM at the somite boundary and MTJ undergoes dynamic changes with changing levels of Fn and increased deposition of Laminins and Collagens (28; 46). Tsp4b interacts with all these ECM components in the MTJ and as a scaffold enables organization of these ECM components enabling muscle attachment at both early and later stages of somite development. The slow muscle fibers, which develop first, initially attach through predominantly integrin dependent interactions with Fn and later as Fn levels decrease, integrin-laminin and integrin-collagen interactions play an important role in attachment. At later stages, dystrophin-dystroglycan adhesion complexes also interact with matrix components of the MTJ (47). Hence, the nature of muscle attachment at the MTJ varies depending on the development stage of muscle fibers and the predominant matrix component involved in the attachment. Tsp4b, as a scaffolding protein and integrin ligand is expressed from early somitogenesis through adult life and modulates the organization of the ECM at the MTJ during every stage of development from somitogenesis.

Hence, the variation in the phenotype between early and later stages of development is probably not due to a change in Tsp4b function but due to the temporal changes in ECM composition at the MTJ. We have emphasized the later role in maintenance because generally muscle attachments form normally in Tsp4b-deficient embryos, but the variable defects we see in muscle attachments and Lam localization at 24-26 hpf could indicate an earlier role. Hence, we have changed our conclusions to state that Tsp4b has a role in both muscle attachment and strengthening/maintenance of ECM at the MTJ.

2a) Discontinuous MTJs: The authors show that Paxillin does not localize to MTJs at 20hpf, there is patchy laminin expression, and Fak phosphorylation is disrupted. The Time Axis needs to be considered. Are the initial somite boundaries forming but not being maintained? Are they not forming? Why are some muscle cells longer than others? And why is this phenotype not reflected in images shown in the beginning? This is a fundamental issue as the phenotype and conclusions drawn change throughout the manuscript. Does a lower dose lead to muscle detachment upon electrostimulation and a higher dose lead to developmental defects?

Initial boundaries form and are maintained for the most part in Tsp4b-deficient embryos – an embryo lacking early Lam or Pxn localization shows a fairly normal somite boundary. Part of the problem in presentation is that MHC staining does not label the entire mediolateral extent of the somites – so not all muscle cells are shown in figures like Figure 1. Similarly, mutations in Itga5 and integrin linked kinase (ilk) do not affect the formation of all somite boundaries and *loc* (ilk^-/-^) mutants muscles also form attachments initially (Koshida et al., 2005; [68]; [58]). Apparently, the early defects we see in Lam and Pxn localization do not prevent muscles from forming attachments other than the occasional mislocalized attachment and an ectopic somite boundary. We have altered the text to clarify that Tsp4b could play roles in both MTJ development and strengthening/maintenance of attachments.

Muscle fibers that are longer than others occur at spots along the somite boundary where they extend across an entire additional somite to form an attachment with the next somite boundary. These could be at locations that lack Lam, in which case individual fibers have to either find the next boundary or form ectopic attachments with neighboring fibers as we sometimes see in Tsp4b-deficient embryos.

We performed our Tsp4b-MO injections as well as our electrostimulation assay in a dose-wise manner to determine optimal voltage (Figure 2—figure supplement 1), and optimal concentrations of MO (Figure 2—figure supplement 1). These chosen conditions have been used in all experiments unless stated otherwise.

3) Lack of integration with the literature: “While the organization of the ECM at myotendinous junctions has been descripted, the developmental processes underlying its establishment and maintenance are poorly studied.” If the purpose of the manuscript is to further understand ECM establishment and maintenance at the zebrafish MTJ a logical approach would be to cogently describe what is known in the fish model. This would include that somites form, what is known about integrin signaling and matrix changes as MTJs form, what ECM proteins are known to be at the fish MTJ, phenotypes when these proteins are disrupted, and pathways that regulate their expression. While the authors are completely correct that MTJ development is not well understood, they should at least establish what is known in the field and how their results contribute. This is also true in the Discussion - multiple labs investigate ECM at these boundaries and have clearly laid out a time course and some cellular and molecular mechanisms of Fn and laminin deposition at these boundaries. Where does tsp4b fit into these pathways?

We have expanded the Introduction and Discussion to add detail regarding the proteins and pathways involved in somite boundary formation and muscle attachment.

*4) The conclusion that pentamerisation of Tsp4 is required for muscle-specific Itg signalling is not well supported (see comments for*
Figure 7
*below) and this should either be toned down or strengthened by further experiment*.

We have performed experiments to verify the effect of the SQAS mutation on Tsp4b structure and function. We have presented the results of this study in Figures 1, 2 and 3 above. A Western blot performed on protein extract from *SQAS-gfp* injected embryos and *tsp4b-FLAG*/*tsp4b-gfp* injected embryos conclusively shows that loss of the cysteine residues prevents Tsp4b to form pentamers. We stained Tsp4b MO-injected embryos injected with SQAS full-length mRNA with anti-Tsp4b and detected no protein at somite boundaries in contrast to Tsp4b-deficient embryos injected with full length wt *tsp4b* mRNA. In order to verify synthesis and secretion of the mutant protein, we made GFP fusions of wt and SQAS mRNAs. Full length *tsp4b*-*gfp* or *SQAS-gfp* mRNAs were co-injected with Tsp4b-MO and imaged live on a confocal microscope at several stages. Both constructs were expressed at early stages of development (tail bud) while by 36 hpf the SQAS-GFP was not localized to somite boundary – there was weak fluorescence throughout the embryo – in contrast to Tsp4b-GFP. SQAS-GFP also disrupted the localization of Tsp4b-FLAG at somite boundaries when co-injected into Tsp4b-deficient embryos. These results conclusively show that CQAC motif in the CCD is required for pentamerization of the protein, which is required for proper localization and function at the somite boundary. We have added the result as Figure 6, Figure 6—figure supplement 1, Figure 6—figure supplement 2, Figure 6—figure supplement 3 and describe the result.

*5) Can the authors explain their choice of mutation from KGD to KGE? Given that KGD appears to substitute for RGD, it is perhaps surprising that KGE fails to substitute for KGD, given that D and E amino acids are closely related*.

Previous studies of RGDs in Fn and other Itg ligands have shown that mutating RGD to RGE abolishes their ability to interact with Itg and promote downstream signaling. Itg interactions with RGD but not RGE are sufficient to promote mTOR activation and S6 protein phosphorylation, which are key steps for adhesion of fibroblasts with Itg ligands (6). Mutation of RGD to RGE in the P2Y2R receptor inhibits its association with Itgs (32). FN^RGE/RGE^ mice show a phenotype similar to that of Itga5/Itgb1-deficient mice (83).

6) Figures

Figure 1—figure supplement 2*: The description here refers to “isolated cells”, but they look like groups of cells in the figure. Please clarify*.

We have replaced “isolated cells” with “groups of cells” in the figure legend.

Figure 5
*and*
Figure 6*: It would be helpful to have the fluorescent images of the MO+RNA-injected embryos, in addition to the controls and the MO only-injected embryos (panels 5I and 6I) and to support the data in panel 5J*.

We have added MO+RNA images to Figure 4 and Figure 5 panels and updated the text and annotations in the legend and main text accordingly.

Figure 7
*legend: The legend for panel B should indicate that these embryos have undergone stimulation, as stated in the text*.

We have added the clause “upon stimulation” in the legend for Figure 7.

Figure 7
*is not convincing. Data for MO-only injected embryos should be shown (as for*
Figure 5*), and rescue should be measured against these embryos rather than uninjected controls. Significance values between the experimental and control samples on this graph are weak, indicating that, in fact, the SQAS RNA can effect a pretty reasonable rescue of both FAK and Laminin levels*.

We have included control + tsp4b-MO injected data in Figure 4 and have provided significance P-value by comparing % of detached muscles with ilk+tsp4b-MO (Figure 4). After consulting with a statistician, we realized that our experimental outcome would require the t-tests to be conducted as “one-tailed” and on making this change to the t-test for the data in Figure 6—figure supplement 3, we obtained a significant P-value <0.05. We have made changes to the figure legend accordingly. In our analyses of Tsp4b-deficient embryos on FAK and Lam localization and subsequent rescue with full length tsp4b mRNA, we have found that the partially rescued FAK and Lam levels were not significantly different from the wt levels. In Figure 6—figure supplement 3, the rescue of Tsp4b-deficient embryos with SQAS full length RNA showed significant change in FAK and Lam levels, suggesting that the rescue was not successful.

Figure 8*: Are these panels showing stimulated muscles, as indicated in the legend, and if so, using what voltage?*

Panels E-H in Figure 7 are embryos that have not been stimulated. Panel I shows the percentage of embryos with detachment upon stimulation. We have changed the legend to clarify these points.